# Sustained expression of HeyL is critical for the proliferation of muscle stem cells in overloaded muscle

**Sumiaki Fukuda[1,2,3], Akihiro Kaneshige[1,2,3], Takayuki Kaji[1], Yu-taro Noguchi[1,3], Yusei Takemoto[1,3], Lidan Zhang[1,3], Kazutake Tsujikawa[3], Hiroki Kokubo[4], Akiyoshi Uezumi[5], Kazumitsu Maehara[6], Akihito Harada[6], Yasuyuki Ohkawa[6], So-ichiro Fukada[1]\***

[1]Project for Muscle Stem Cell Biology, Graduate School of Pharmaceutical Sciences, Osaka University, Suita, Japan; [2]Biological/Pharmacological Research Laboratories, Central Pharmaceutical Research Institute, Japan Tobacco Inc, Takatsuki, Japan; [3]Laboratory of Molecular and Cellular Physiology, Graduate School of Pharmaceutical Sciences, Osaka University, Suita, Japan; [4]Department of Cardiovascular Physiology and Medicine, Graduate School of Biomedical and Health Sciences, Hiroshima University, Hiroshima, Japan; [5]Muscle Aging and Regenerative Medicine, Research Team for Geriatric Medicine, Tokyo Metropolitan Institute of Gerontology, Tokyo, Japan; [6]Division of Transcriptomics, Medical Institute of Bioregulation, Kyushu University, Fukuoka, Japan

**Abstract** In overloaded and regenerating muscle, the generation of new myonuclei depends on muscle satellite cells (MuSCs). Because MuSC behaviors in these two environments have not been considered separately, MuSC behaviors in overloaded muscle remain unexamined. Here, we show that most MuSCs in overloaded muscle, unlike MuSCs in regenerating muscle, proliferate in the absence of MyoD expression. Mechanistically, MuSCs in overloaded muscle sustain the expression of *Heyl*, a Notch effector gene, to suppress MyoD expression, which allows effective MuSC proliferation on myofibers and beneath the basal lamina. Although *Heyl*-knockout mice show no impairment in an injury model, in a hypertrophy model, their muscles harbor fewer new MuSC-derived myonuclei due to increased MyoD expression and diminished proliferation, which ultimately causes blunted hypertrophy. Our results show that sustained HeyL expression is critical for MuSC proliferation specifically in overloaded muscle, and thus indicate that the MuSC-proliferation mechanism differs in overloaded and regenerating muscle.
DOI: https://doi.org/10.7554/eLife.48284.001

\*For correspondence:
fukada@phs.osaka-u.ac.jp

## Introduction

Skeletal muscle consists of multinuclear cells named myofibers and exhibits the ability to regenerate. When myofibers are damaged, mitotically quiescent mononuclear cells known as muscle satellite (stem) cells (MuSCs) exit quiescence and become activated and immediately start expressing MyoD (a myogenic determination gene), followed by entry into the cell cycle to yield daughter cells (myoblasts) (*Relaix and Zammit, 2012*). To generate multinuclear cells, the proliferated myoblasts fuse with each other or with existing myofibers through the action of the fusion proteins 'myomaker' and 'myomixer/myomerger/minion' (*Bi et al., 2017*; *Millay et al., 2013*; *Quinn et al., 2017*; *Zhang et al., 2017*). The indispensable role and behaviors of MuSCs in muscle regeneration have

been comprehensively investigated, leading to the establishment of an MuSC-differentiation model (*Lepper et al., 2011*; *Sambasivan et al., 2011*; *Wang and Rudnicki, 2011*).

Another ability possessed by skeletal muscle is the capacity to adapt its size in response to new milieus. A loss of muscle mass (i.e., muscle atrophy) occurs in bedridden people, patients with cancer cachexia or sepsis, or during aging, whereas an increase in muscle mass (i.e., muscle hypertrophy) is induced by resistance training. Although MuSC requirement for myofiber hypertrophy has been debated, MuSCs are required for long-term muscle hypertrophy (*Egner et al., 2016*; *Fry et al., 2014*; *Fukada, 2018*; *Goh and Millay, 2017*). Furthermore, MuSCs are indispensable for supplying new nuclei to myofibers during overload-induced muscle hypertrophy (*Egner et al., 2016*; *Goh and Millay, 2017*; *McCarthy et al., 2011*; *Schiaffino et al., 1976*). Similar to the regeneration process, myomaker is necessary for MuSC fusion with myofibers during muscle hypertrophy. *Goh and Millay (2017)* showed that the addition of new myonuclei derived from MuSCs was severely hampered in overloaded muscle from myomaker-deficient mice. Before this cell fusion, MuSCs must escape from their quiescent state, become activated, and proliferate; however, compared with MuSC behavior during muscle regeneration, MuSC-activation/proliferation mechanisms during hypertrophy remain poorly investigated. One reason for this is that damage to skeletal muscle arises from resistance training (*Clarkson and Hubal, 2002*; *Ebbeling and Clarkson, 1989*; *Kuipers, 1994*); therefore, previous studies have not considered whether hypertrophic muscle requires special mechanisms for MuSC activation/proliferation.

Specific forms of exercise (e.g., eccentric contraction) can result in disruption of the myofibrillar structure and leakiness of the myofiber membrane (sarcolemma) (*Fridén et al., 1983*; *Grounds, 1998*). However, these disruptions depend on the intensity, duration, and frequency of the administered exercise and do not unfailingly lead to lethal/substantial injury to the myofiber (*Grounds, 1998*). Intriguingly, studies conducted by Darr and Schultz suggested that MuSCs became activated and replicated, even on myofibers not overtly necrotic in eccentrically exercised soleus and extensor digitorum longus muscles of rats. This finding suggests that the existence of lethal damage to myofibers is not necessary for MuSC activation and proliferation in overloaded muscle. Given that this mechanism underlying MuSC activation/proliferation in overloaded muscle differs from that in regenerating muscle, elucidation of MuSC behaviors in overloaded muscle could facilitate the development of new therapeutic strategies for the muscle wasting or atrophy that occurs in the absence of muscle damage (as in chronic diseases and aging).

Under steady state conditions, MuSCs are maintained in an undifferentiated and quiescent state in their niche (*Fukada, 2018*), with the molecules and pathways that maintain MuSCs in the dormant state having been recently revealed (*Baghdadi et al., 2018*; *Cheung and Rando, 2013*). Among the pathways involved, the canonical Notch signaling pathway is crucial for sustaining the undifferentiated state in MuSCs. Notch signaling is an evolutionarily conserved intercellular signaling system required for cell-fate decisions and patterning events (*Lai, 2004*). The most widely recognized primary targets of canonical Notch signaling are the (hairy and enhancer of split) *Hes* and (Hes-related; also known as Hesr/Herp/Hrt/Gridlock/Chf) *Hey* families of bHLH transcriptional-repressor genes (*Iso et al., 2003*), which function to suppress MyoD and myogenin expression in MuSCs. Recently, Lahmann et al. indicated that Hes1 controls oscillatory *Myod* expression in activated/proliferating MuSCs (*Lahmann et al., 2019*). We previously reported that neither *Hey1* (Hey1-KO) nor *Heyl* single-knockout (HeyL-KO) mice show abnormalities in regenerative ability, MyoD and myogenin expression, or MuSC number; however, double-KO mice exhibit severe regenerative defects due to a reduction in MuSC number resulting from increased MyoD and myogenin expression in the MuSCs (*Fukada et al., 2011*; *Noguchi et al., 2019*). When MuSCs respond to muscle injury, *Hey1* and *Heyl* expression levels are drastically decreased; therefore, activation/proliferation of MuSCs in injured muscle do not require *Hey1* and *Heyl* expression. However, *Hey1* and *Heyl* expression and function in proliferating MuSCs in hypertrophic muscle remain unknown.

To investigate the mechanism of MuSC activation/proliferation during muscle hypertrophy, we first examined the protein-expression patterns of MyoD and Ki67 in MuSCs in overloaded muscle. During MuSC activation in acute injury or in vitro culture, almost all MuSCs initially express MyoD at the protein level and then express Ki67 (*Ogawa et al., 2015*). However, we found that in overloaded muscle, the majority of MuSCs proliferated in the absence of MyoD protein expression. Notably, gene-expression analyses indicated that *Hey1* expression in proliferating MuSCs was decreased to the same extent as that observed in the injury model, whereas *Heyl* expression was sustained in

overloaded MuSCs, suggesting that the requirement of HeyL in MuSCs differed between regenerating and overloaded muscle. Accordingly, HeyL-KO mice showed no marked defects in a muscle-injury model but exhibited decreases in the number of proliferating MuSCs, MyoD-negative cells, and incorporated myonuclei in overloaded muscle, indicating that sustained HeyL expression is critical for MuSC expansion in overloaded muscle. Finally, both muscle weight and myofiber size were attenuated in the overloaded muscle of HeyL-KO mice at 9 weeks after synergist ablation. Collectively, our findings indicate that sustained HeyL expression is necessary for effective proliferation of MuSCs in overloaded muscle, and that the MuSC-proliferation mechanism in overloaded muscle differs from that in injured muscle.

## Results

### Proliferation of MuSCs during muscle hypertrophy

To elucidate the molecular mechanisms by which MuSCs become activated and proliferate in overloaded muscle, the Achilles tendon in mice was cut to induce compensatory hypertrophy in plantaris muscle (tenotomy). This model relies on the compensatory adaptation of the plantaris muscle following cutting of the functionally synergistic muscles of the Achilles tendon (i.e., the gastrocnemius and soleus muscles). Surgical ablation of gastrocnemius and soleus muscles [synergist ablation; SA] leads to more marked increase in muscle weight than after tenotomy alone; however, the incidence of muscle damage is also increased. The rate of increase in muscle weight in the tenotomy model was lower than that in the SA model (*Figure 1A*), although muscle weight increased significantly (*Figure 1B*). Moreover, areas of muscle damage were rarely detected (*Figure 1C*; *Figure 1—figure supplement 1*). Notably, even under conditions of diminished muscle damage, we observed an increased number of Pax7$^+$ cells (*Figure 1D*). Accordingly, we frequently observed cells positive for Ki67 (a proliferation marker) among Pax7$^+$ cells (*Figure 1E*), and importantly, at 4 days after tenotomy, the percentage of Pax7$^+$Ki67$^+$ cells among Pax7$^+$ cells (~70%) was higher than that of Pax7$^+$-MyoD$^+$ cells (~35%), suggesting that at least >50% of these MuSCs proliferated in the absence of MyoD expression (*Figure 1E,F*).

To further examine the expression pattern of MyoD and Ki67 in Pax7$^+$ cells in overloaded muscle, we isolated single myofibers from overloaded plantaris muscle and immunostained them concurrently for MyoD and Ki67 (*Figure 2A*) and used 3 day cultured single myofibers isolated from intact skeletal muscle as a positive control (*Figure 2A*). MyoD and Ki67 were expressed by almost all Pax7$^+$ cells on the cultured myofibers (*Figure 2B*), with these results similar to those reported previously using an in vitro culture model and in vivo regenerating muscle (*Ogawa et al., 2015*; *Zammit et al., 2004*). By contrast, the majority (~70%) of activated Pax7$^+$ cells on overloaded myofibers were positive for Ki67 but negative for MyoD (*Figure 2B,C*), which agreed with our immunohistochemical analysis (*Figure 1E,F*). In sham-operated muscle, 93% of Pax7$^+$ cells on myofibers expressed neither MyoD nor Ki67, indicating that these cells were in a quiescent state (*Figure 2B,C*).

Our results showed that few areas of muscle damage were present in our overload model (*Figure 1C*); however, factors released from locally damaged myofibers could have affected MuSC activation/proliferation in the overloaded muscle. In this scenario, MuSCs would proliferate in a restricted area and on a few myofibers. Conversely, if MuSCs respond to other stimuli (e.g., mechanical stimulation), MuSCs on almost all myofibers would escape from the quiescent state and proliferate on undamaged myofibers. To investigate this, we determined both the number and properties of Pax7$^+$ cells on single myofibers by using isolated single myofibers in the assays. Because damaged myofibers could be lost during the preparation of single myofibers due to their hyper-contraction, the observed Pax7$^+$ cells were considered to be present on undamaged myofibers. Under this experimental condition, the number of Pax7$^+$ cells per myofiber was increased overall in the overloaded muscles but not in only limited areas of the myofibers (*Figure 2D*). We then examined the properties of Pax7$^+$ cells on single myofibers. In this analysis, if all five Pax7$^+$ cells present on a myofiber were found to be in the quiescent state (MyoD$^-$Ki67$^-$), they were categorized into 100% in *Figure 2E* and 0% in *Figure 2F*, respectively. However, if zero to one of the five Pax7$^+$ cells on the myofiber were in the quiescent state, the cells were categorized into the 0% to 20% range in *Figure 2E*, whereas if three to four of the five Pax7$^+$ cells were MyoD$^-$Ki67$^+$, they were categorized into the 60% to 80% range in *Figure 2F*. As expected, almost all sham-operated myofibers harbored

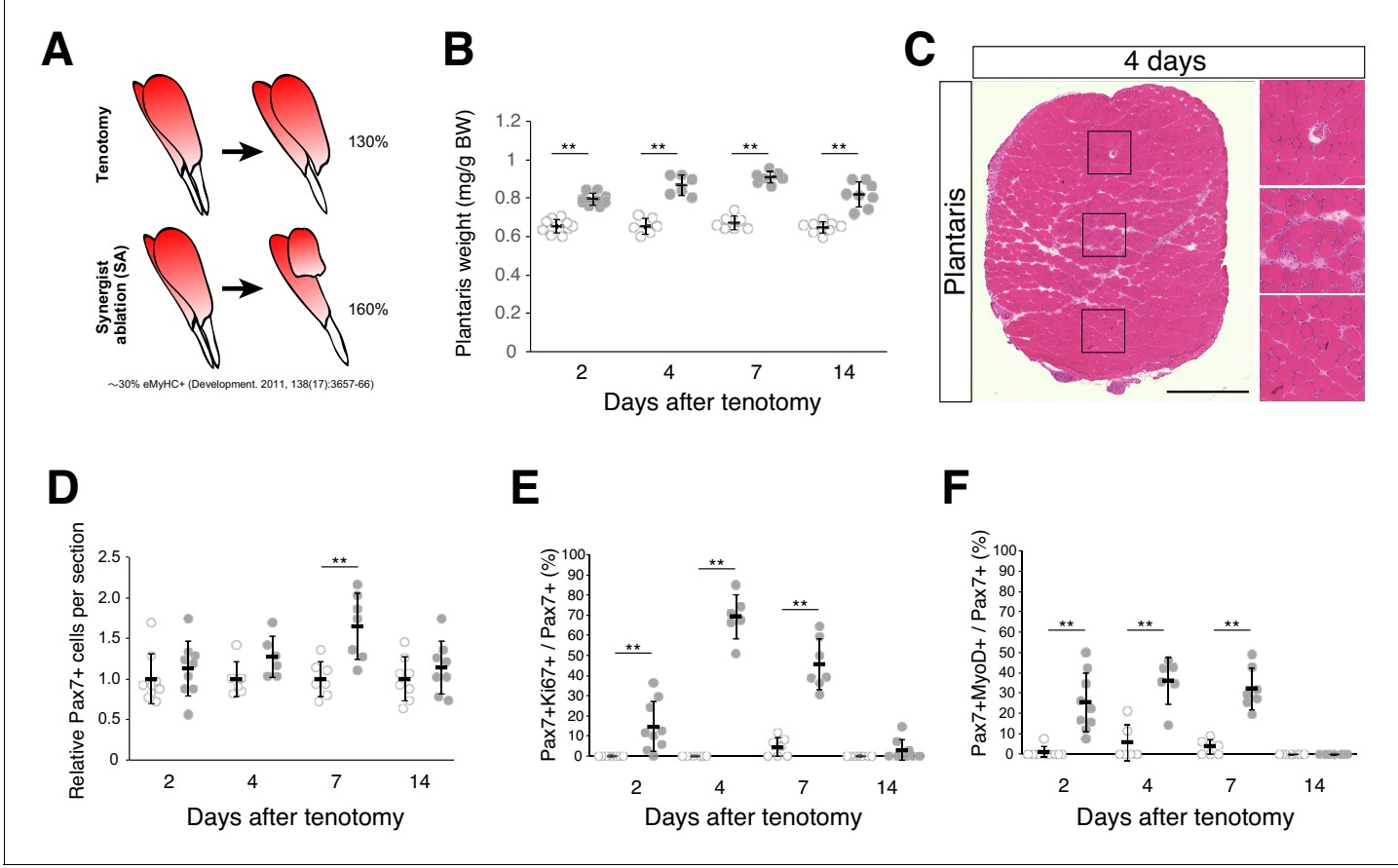

**Figure 1.** Tenotomy induces activation and proliferation of muscle stem cells in the absence of substantial muscle damage. (**A**) Two mouse models that induce muscle hypertrophy. (**B**) Plantaris muscle weight relative to body weight (mg/g) after tenotomy (gray circle) or the weight of contralateral sham-operated muscle (open circle). **p<0.01. (**C**) H and E staining of plantaris muscle sections at 4 days after tenotomy. Scale bar: 500 μm. Right three images in squares represent magnifications of the left image. (**D**) Relative number of MuSCs in plantaris muscle sections after tenotomy (gray circle) or in contralateral sham muscle sections (open circle). **p<0.01. (**E**) Percentage of Ki67+ cells among Pax7+ cells in contralateral sham (open circle) or tenotomy induced plantaris muscle (gray circle). **p<0.01. (**F**) Percentage of MyoD+ cells among Pax7+ cells in contralateral sham (open circle) or tenotomy induced plantaris muscle (gray circle). **p<0.01. Data represent the means ± standard deviation. A two-sided Student's *t* test or Welch's *t* test was performed.

DOI: https://doi.org/10.7554/eLife.48284.002

The following source data and figure supplement are available for figure 1:

**Source data 1.** Numerical data for making the graphs in *Figure 1*.
DOI: https://doi.org/10.7554/eLife.48284.004
**Figure supplement 1.** Relative absence of damaged myofibers in plantaris muscle after tenotomy.
DOI: https://doi.org/10.7554/eLife.48284.003

only quiescent MuSCs (MyoD⁻Ki67⁻). By contrast,>80% of overloaded myofibers harbored <40% quiescent MuSCs (*Figure 2E*), and ~60% of the myofibers harbored >40% MyoD⁻Ki67⁺ cells (*Figure 2F*). These results suggested that escape of MuSCs from quiescent state did not occur in a restricted area of the overloaded myofibers, and that almost all myofibers responded to the overload. Collectively, these data suggest that MuSCs proliferate during hypertrophy in a manner distinct from that during muscle injury.

### *Heyl* expression is sustained in proliferating MuSCs in overloaded muscle

In contrast to what occurs during the regenerative process, myofibers are retained in overloaded muscle. Accordingly, we observed proliferating MuSCs between myofibers and the basal lamina in overloaded myofibers (*Figure 3A,B*). Canonical Notch signaling is a critical pathway for suppressing

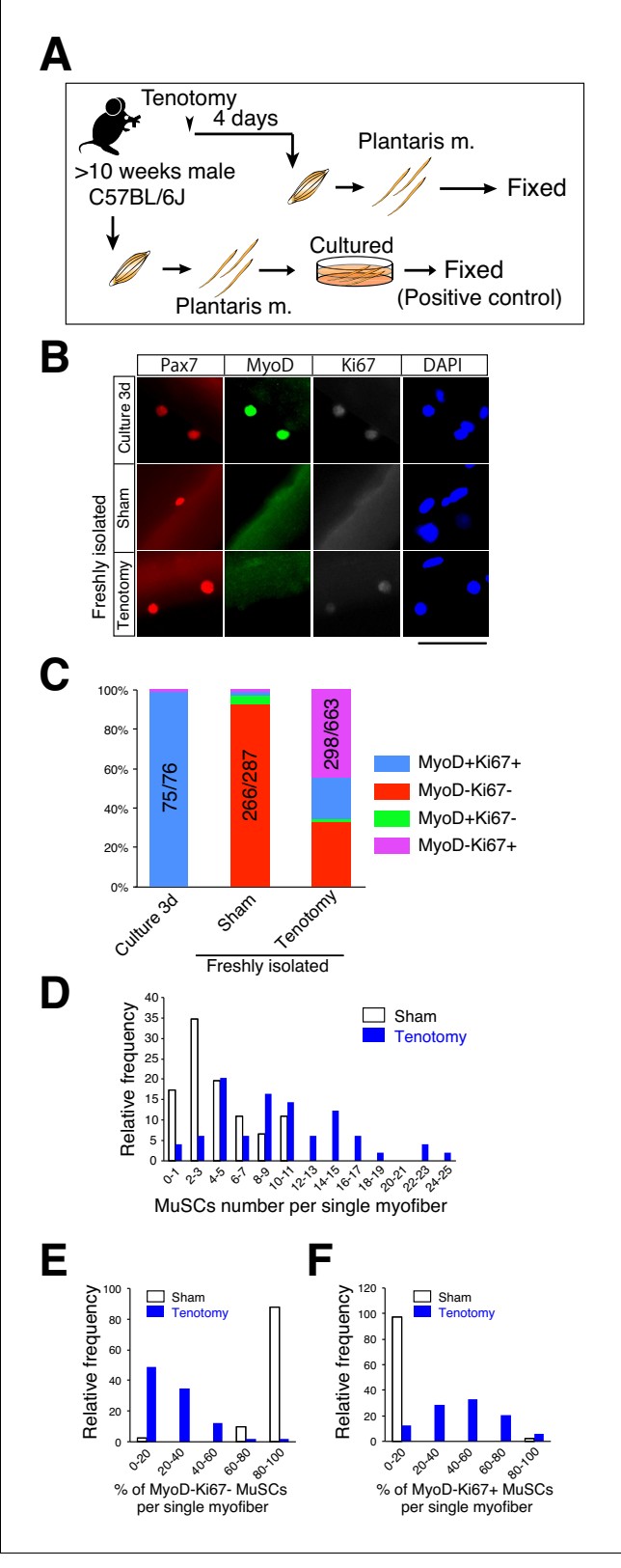

**Figure 2.** Majority of muscle stem cells proliferate on undamaged overloaded myofibers in the absence of MyoD expression. (**A**) Experimental scheme for analyzing the number and properties of MuSCs on isolated single myofibers. At 4 days after tenotomy, single myofibers were isolated and immediately fixed. Single myofibers cultured for 3 days were used as positive controls for immunostaining. (**B**) Immunostaining of Pax7 (red), MyoD

*Figure 2 continued on next page*

*Figure 2 continued*

(green), and Ki67 (white) in MuSCs on 3 day cultured myofibers (Culture 3d; positive control), freshly isolated myofibers of sham muscle (Sham), or freshly isolated myofibers at 4 days after tenotomy (Tenotomy). Nuclei were counterstained with DAPI. Scale bar: 50 µm. (C) Bar graph showing frequencies of MyoD$^+$Ki67$^+$ (blue column), MyoD$^-$Ki67$^-$ (red column), MyoD$^+$Ki67$^-$ (green column), and MyoD$^-$Ki67$^+$ (purple column) cells on 3 day cultured myofibers, sham myofibers, or myofibers at 4 days after tenotomy. The number associated with the majority phenotype in Pax7$^+$ cells is indicated in the numerator of each bar. Total number of Pax7$^+$ cells is indicated in the denominator of each bar. (D) Histogram showing the frequency of the listed number of MuSCs per myofiber in the case of sham myofibers (white bar, $n = 46$) or myofibers at 4 days after tenotomy (blue bar, $n = 49$). The X-axis indicates the number of MuSCs per myofiber, and the Y-axis shows the frequency of myofibers harboring the indicated number of MuSCs. (E, F) Frequency of MyoD$^-$Ki67$^-$ (E) or MyoD$^-$Ki67$^+$ (F) MuSCs on single myofibers: sham control (white bar, $n = 41$) versus 4 days after tenotomy (blue bar, $n = 49$). The X-axis indicates the percentage of MyoD$^-$Ki67$^-$ or MyoD$^-$Ki67$^+$ MuSCs per myofiber. The Y-axis shows the frequency of myofibers with the indicated percentage of MyoD$^-$Ki67$^-$ (E) or MyoD$^-$Ki67$^+$ (F) MuSCs per myofiber.
DOI: https://doi.org/10.7554/eLife.48284.005
The following source data is available for figure 2:

**Source data 1.** Numerical data for making the graphs in *Figure 2*.
DOI: https://doi.org/10.7554/eLife.48284.006

MyoD expression in MuSCs (*Bjornson et al., 2012*; *Mourikis et al., 2012*), and myofibers are considered a source of Notch ligand. Therefore, we hypothesized that canonical Notch signaling is responsible for MyoD suppression in proliferating MuSCs in overloaded muscle. *Hey1*, *Heyl,* and *Hes1* are crucial primary targets of canonical Notch signaling in MuSCs (*Fukada et al., 2011*; *Lahmann et al., 2019*) and function to suppress myogenic differentiation as Hey–Hes1 heterodimer complexes (*Noguchi et al., 2019*). Moreover, *Hey1* and *Heyl* are both expressed in quiescent MuSCs, although their expression is drastically reduced during injury induced MuSC activation (*Figure 3—figure supplement 1*) (*Fukada et al., 2011*; *Mourikis et al., 2012*). Although neither *Hey1* (Hey1-KO) nor *Heyl* single-KO (HeyL-KO) mice show impaired muscle regeneration, *Hey1/Heyl* double-KO mice exhibit increased MyoD expression in MuSCs, indicating redundant roles of Hey1 and HeyL in MuSCs (*Fukada et al., 2011*; *Noguchi et al., 2019*). To determine the expression of Notch target genes, we performed RNA-seq analyses and found that their expression was overall sustained in overloaded MuSCs at 4 days after tenotomy (*Figure 3C*). Specifically, we observed significant sustained expression of *Heyl* in overloaded MuSCs as compared with that observed in regenerating MuSCs (*Figure 3—figure supplement 1*). We then confirmed the RNA-seq results by quantitative reverse transcription polymerase chain reaction (qRT-PCR), with the data showing that *Heyl* but not *Hey1* expression was sustained in tenotomy induced MuSCs (*Figure 3D*). Additionally, we measured levels of *Col5a1*, *Col5a3*, and *Col6a1* expression as newly identified direct target genes of Rbp-J (the major transcriptional effector of Notch signaling) in MuSCs (*Baghdadi et al., 2018*). We found that their expression levels were reduced at 2 days after surgery but comparable to levels in quiescent MuSCs at 4 days after surgery, when MuSCs exhibit active proliferation in overloaded muscle (*Figure 3E*). Moreover, RNA-seq data indicated increased expression levels of cell cycle-related genes and sustained expression of *Myf5* in overloaded MuSCs (*Figure 3B*). These findings suggested that MuSCs proliferate under activation of canonical Notch signaling in the overload mouse model, and that sustained *Heyl* expression is a feature of proliferating MuSCs in overloaded muscle.

## Proliferation ability of MuSCs in HeyL-KO mice

We then investigated the role of sustained *Heyl* expression in MuSC behavior in overloaded muscle by inducing tenotomy in HeyL-KO mice and comparing the effects with those in Hey1-KO and control mice (*Figure 4A*). Interestingly, in HeyL-KO mice, the frequency of Ki67$^+$MyoD$^+$ and Ki67$^+$MyoD$^-$ cells among Pax7$^+$ cells increased and decreased, respectively, at 4 days after surgery (*Figure 4B,C*); therefore, the ratio of MyoD$^+$ to MyoD$^-$ cells among proliferating cells (Ki67$^+$Pax7$^+$) was markedly higher in HeyL-KO mice than in control and Hey1-KO mice (*Figure 4D*). Moreover, the frequency of Ki67$^+$ cells in HeyL-KO mice was similar to that in control mice, whereas the total number of Pax7$^+$ cells per myofiber at 4 days after surgery was significantly lower in HeyL-KO mice than in control and Hey1-KO mice (*Figure 4E*), suggesting that the loss of *Heyl* affected MuSC

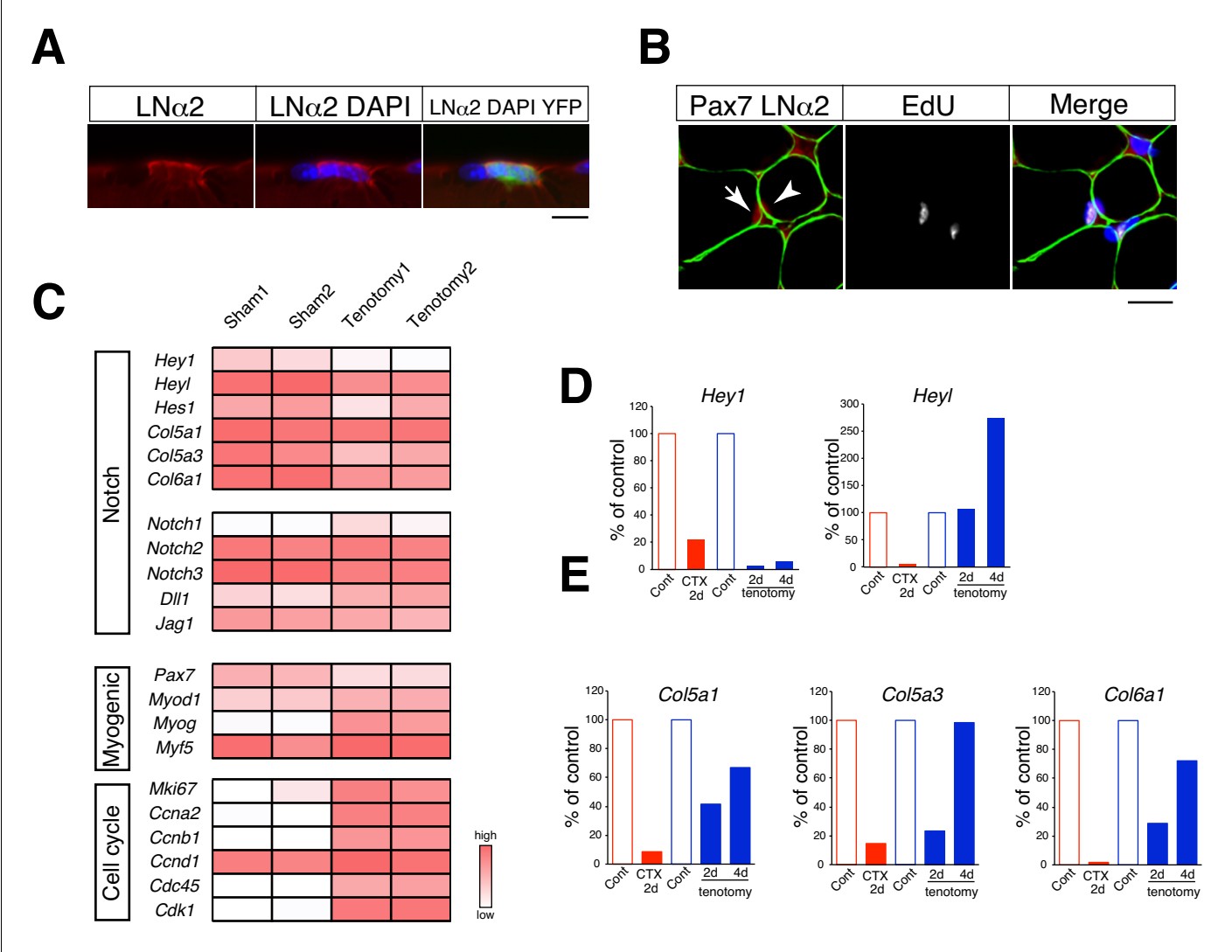

**Figure 3.** Relative mRNA expression of Notch target genes in muscle stem cells in regenerating versus overloaded muscle. (**A**) Immunostaining of laminin α2 (LNα2; red) and YFP (green) in isolated myofibers derived from *Pax7^CreERT2+^::Rosa^YFP^* (Rosa-YFP) mice at 4 days after tenotomy. Nuclei were counterstained with DAPI. Scale bar: 10 μm. (**B**) Immunostaining of Pax7 (red), laminin α2 (LNα2; green), and EdU (white) in sections from *C57BL/6* mice at 4 days after tenotomy. Nuclei were counterstained with DAPI. Arrow or arrowhead indicate EdU⁺ or EdU⁻ Pax7⁺ cells. Scale bar: 20 μm. (**C**) RNA-seq analyses of Notch, myogenic, and cell-proliferation-related genes in MuSCs from sham or overloaded plantaris muscle of Rosa-YFP at 4 days after tenotomy. (**D, E**) Mononuclear myogenic cells were isolated from intact muscle (Cont; red), damaged muscle at 2 days after cardiotoxin injection (CTX 2d), sham-operated muscle (Cont; blue), or overloaded plantaris muscle of C57BL/6. Relative expression of *Hey1* and *Heyl* (**D**) and three other canonical Notch target genes (*Col5a1*, *Col5a3*, and *Col6a1*) were compared between Cont and CTX/tenotomy samples (**E**).

DOI: https://doi.org/10.7554/eLife.48284.007

The following figure supplement is available for figure 3:

**Figure supplement 1.** Expression of Notch, myogenic, and cell-proliferation-related genes in MuSCs from intact or damaged muscles.
DOI: https://doi.org/10.7554/eLife.48284.008

proliferation but not their activation. The results at 7 days after surgery were similar to those at 4 days after surgery (*Figure 4—figure supplement 1*). By contrast, in vivo regeneration and in vitro culture assays revealed no notable differences between control and HeyL-KO mice (*Figure 4—figure supplements 2* and *3*). Collectively, these results indicate that the sustained expression of *Heyl* in MuSCs is critical for suppressing MyoD expression in overloaded muscles, and that the loss of *Heyl* impairs the proliferation of MuSCs by biasing them toward differentiation (MyoD expression).

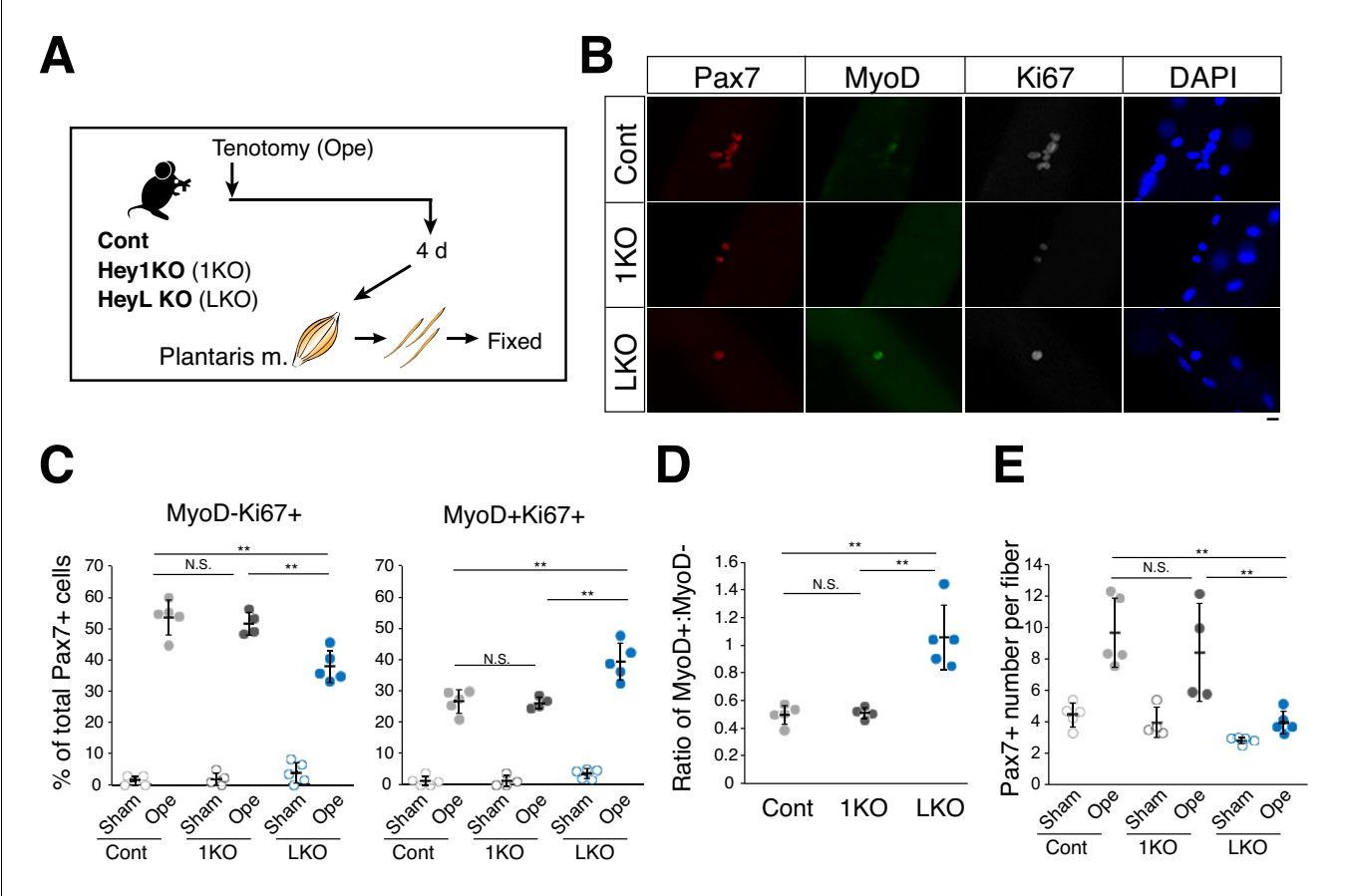

**Figure 4.** Increased ratio of MyoD-expressing myogenic cells and decreased cell number in overloaded HeyL-KO mice. (A) Experimental scheme for analyzing Pax7+ cells on plantaris myofibers derived from control (Cont), Hey1-KO (1KO), or HeyL-KO (LKO) mice at 4 days after tenotomy (Ope). (B) Immunostaining of Pax7 (red), MyoD (green), and Ki67 (white) in MuSCs on freshly isolated myofibers at 4 days after tenotomy from Cont, 1KO, or LKO mice. Nuclei were counterstained with DAPI. Scale bar: 20 μm. (C) Frequency of MyoD−Ki67+ (left) or MyoD+Ki67+ (right) cells among Pax7+ cells on myofibers: sham (open circle) versus 4 days after tenotomy (closed circle), derived from Cont (light gray, $n = 5$), 1KO (black, $n = 4$), or LKO (blue, $n = 5$) mice. **$p<0.01$. N.S.: not significant. (D) Ratio of MyoD+ to MyoD− cells among Ki67+Pax7+ cells on myofibers at 4 days after tenotomy, derived from Cont (light gray, $n = 5$), 1KO (black, $n = 4$), or LKO (blue, $n = 5$) mice. **$p<0.01$. N.S.: not significant. (E) Number of Pax7+ cells on myofibers: sham (open circle) versus 4 days after tenotomy (closed circle), derived from Cont (light gray, $n = 5$), 1KO (black, $n = 4$), or LKO (blue, $n = 5$) mice. **$p<0.01$. N.S.: not significant. Data represent the mean ± standard deviation. One-way ANOVA, followed by the Tukey–Kramer test, was used for statistical analyses.

DOI: https://doi.org/10.7554/eLife.48284.009

The following source data and figure supplements are available for figure 4:

**Source data 1.** Numerical data for making the graphs in *Figure 4*.
DOI: https://doi.org/10.7554/eLife.48284.016

**Figure supplement 1.** Increased ratio of MyoD-expressing myogenic cells and decreased cell number in overloaded HeyL-KO mice at 7 days after tenotomy.
DOI: https://doi.org/10.7554/eLife.48284.010

**Figure supplement 1—source data 1.** Numerical data for making the graphs in *Figure 4—figure supplement 1*.
DOI: https://doi.org/10.7554/eLife.48284.011

**Figure supplement 2.** Normal muscle-regenerative ability in HeyL-KO mice.
DOI: https://doi.org/10.7554/eLife.48284.012

**Figure supplement 2—source data 1.** Numerical data for making the graphs in *Figure 4—figure supplement 2*.
DOI: https://doi.org/10.7554/eLife.48284.013

**Figure supplement 3.** Normal activation and proliferation ability of HeyL-KO MuSCs on cultured myofibers.
DOI: https://doi.org/10.7554/eLife.48284.014

**Figure supplement 3—source data 1.** Numerical data for making the graphs in *Figure 4—figure supplement 3*.
DOI: https://doi.org/10.7554/eLife.48284.015

## The supply of new myonuclei is diminished in overloaded muscle in HeyL-KO mice

We then compared the numbers of new myonuclei supplied by MuSCs in overloaded muscle from control, Hey1-, and HeyL-KO mice. Proliferating MuSCs were labeled with EdU (*Figure 5A*), and MuSC-derived nuclei (EdU$^+$ myonuclei) were discriminated from other nuclei after staining for dystrophin [i.e., we detected new MuSC-derived myonuclei as EdU$^+$ nuclei located beneath the myofiber membrane (marked by dystrophin staining)] (*Fukada et al., 2011*) (*Figure 5B*). Approximately 20 or 60 myonuclei were labeled per cross-section from control/Hey1-KO mice at 4 or 7 days after surgery, respectively (*Figure 5C*), which suggested that the proliferating MuSCs actively fused with myofibers during this period. Notably, in HeyL-KO mice, we detected <50% as many new (EdU$^+$) myonuclei per one cross-section or per 100 myofibers as compared with that detected in control and Hey1-KO mice at 7 days after surgery (*Figure 5C*; *Figure 5—figure supplement 1A*). Additionally, the number of myonuclei per 100 myofibers in HeyL-KO mice after tenotomy was lower than that in control mice (*Figure 5—figure supplement 1B*). Furthermore, in sham-operated muscle, EdU$^+$ myonuclei were rarely detected in control, Hey1-, and HeyL-KO mice (*Figure 5C*), which indicated that changes in the number of EdU$^+$ myonuclei were due to MuSC fusion with myofibers in response to overload. However, similar to a previous MuSC-depletion study (*McCarthy et al., 2011*), the decrease in the number of new myonuclei in HeyL-KO mice did not correlate with muscle weight at this time point (*Figure 5D*). These findings indicate that the impaired expansion of MuSCs (due to increased MyoD expression) resulted in decreased new myonuclei in the overloaded muscle of HeyL-KO mice.

## HeyL loss affects muscle weight and myofiber size in SA-induced hypertrophy

In contrast to tenotomy, a previous study reported that SA allows the plantaris muscle to continue increasing in size, and that the effects of diminished MuSC number and function were observed at a later stage of SA-induced hypertrophy (*Fry et al., 2014*). To examine whether the decreased number of new myonuclei derived from MuSCs in HeyL-KO mice affects muscle weight and myofiber size, we induced SA in control and HeyL-KO mice (*Figure 6A*). Similar to the previous study, the muscle weight to body weight ratio increased by up to 190% relative to that in sham contralateral muscle in control mice (sham vs. SA: 0.65 ± 0.04 vs. 1.26 ± 0.20 mg/g) (*Figure 6B*) (*Fry et al., 2014*). Moreover, histologic analyses of plantaris muscle revealed an increased myofiber cross-sectional area in control mice (sham vs. SA: 1127 ± 227 vs. 1794 ± 349 μm$^2$) (*Figure 6C,D*); however, in HeyL-KO mice, the growth rate of plantaris muscle was lower than that in control mice (HeyL-KO sham vs. SA: 0.63 ± 0.05 vs. 1.07 ± 0.13 mg/g) (*Figure 6B*), and after SA, myofibers in the KO mice were also smaller than those in control mice (sham vs. SA: 1116 ± 126 vs. 1443 ± 205 μm$^2$) (*Figure 6C,D*). Furthermore, HeyL-KO mice following SA tended to have fewer myonuclei compared with control mice (*Figure 6—figure supplement 1*). These findings indicate that the loss of *Heyl* in MuSCs results in blunted muscle hypertrophy at the late stage in SA-treated mice.

## Discussion

One of the most crucial findings obtained in this study was that MuSCs proliferated in the absence of MyoD expression by expressing HeyL in overloaded muscle (*Figure 7A*), unlike findings in well-known myogenic differentiation models. Therefore, the mechanism of MuSC proliferation in overloaded muscle differs from those described in regenerating muscle and in vitro culture systems, which have provided the basis for concepts of MuSC proliferation and differentiation (*Fukada, 2018*; *Zammit et al., 2004*; *Zammit et al., 2006*). Here, we showed that in overloaded muscle, MuSCs proliferated between the basal lamina and myofibers. By contrast, in regenerating muscle, MuSCs proliferate beneath the retained basal lamina ('ghost fibers'), where damaged myofibers have been removed by macrophages (*Webster et al., 2016*). Single-myofiber culture does not allow MuSCs to proliferate beneath the basal lamina, because MuSCs escape from the niche and proliferate outside the basal lamina (*Siegel et al., 2009*; *Yamaguchi et al., 2015*). The environment in which MuSCs proliferate is considered to be responsible for the difference in MyoD- and HeyL-expression patterns in MuSCs between regenerating and overloaded muscle.

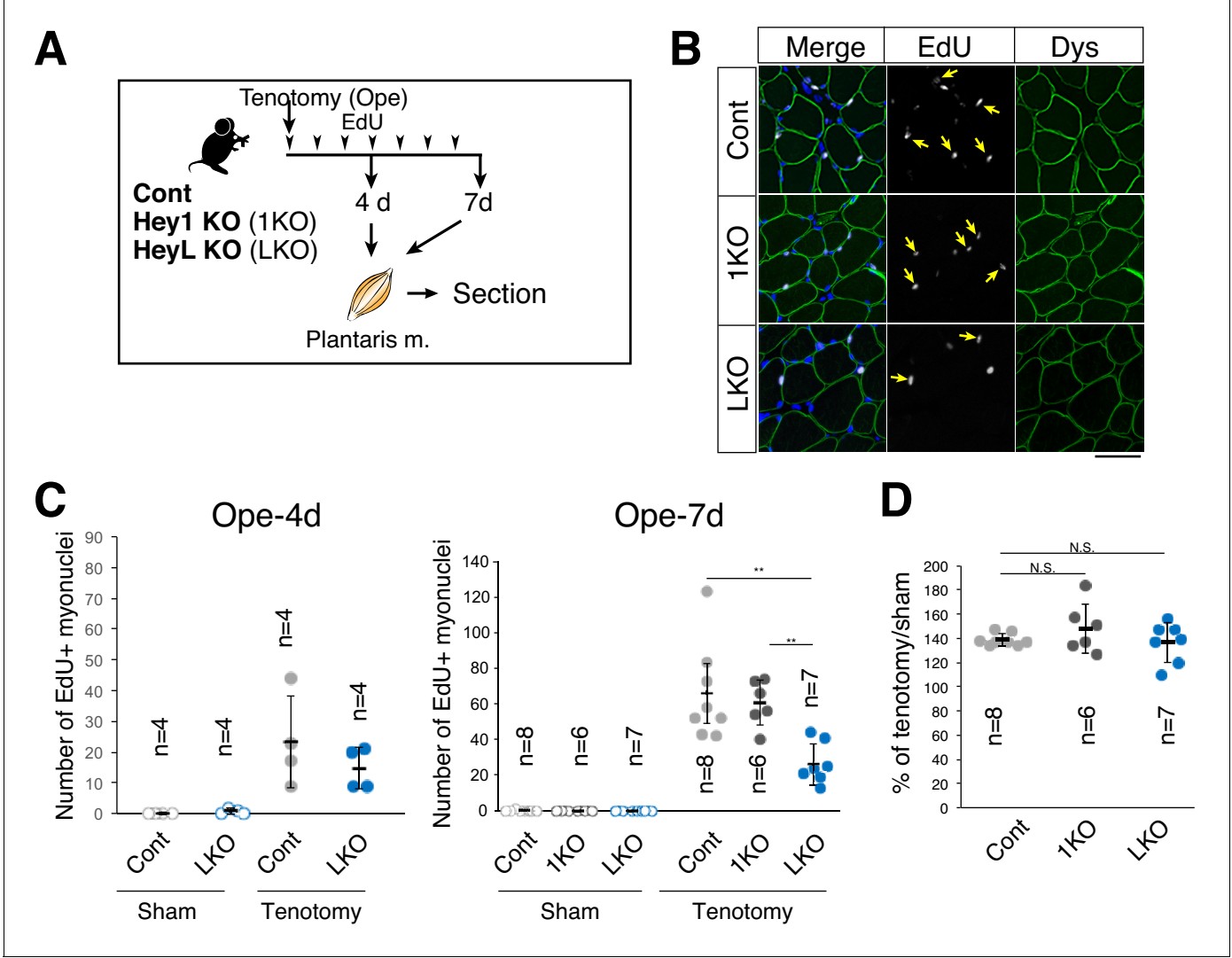

**Figure 5.** Decreased number of new myonuclei supplied by muscle stem cells in overloaded HeyL-KO muscle. (A) Experimental scheme for analyzing new MuSC-derived myonuclei in plantaris muscle of control (Cont), Hey1-KO (1KO), or HeyL-KO (LKO) mice at 4 or 7 days after tenotomy. Arrowheads indicate the day of EdU injection. (B) Arrows indicate new myonuclei supplied by MuSCs in overloaded muscle of Cont, 1KO, or LKO mice. MuSC-derived myonuclei were labeled with EdU (white) and located inside the staining for dystrophin (Dys; green). Nuclei were counterstained with DAPI. Scale bar: 50 μm. (C) Number of EdU$^+$ myonuclei in muscle: sham (open circle) versus tenotomy (closed circle). Left graph: results at 4 days after tenotomy (Ope-4d) (Cont: light gray, $n = 4$; LKO: blue, $n = 4$). Right graph: results at 7 days after tenotomy (Cont: light gray, $n = 8$; 1KO: black, $n = 6$; LKO: blue, $n = 7$). **$p<0.01$. N.S.: not significant. (D) Ratio of overloaded muscle weight to sham muscle weight in Cont (light gray, $n = 8$), 1KO (black, $n = 6$), or LKO (blue, $n = 8$) mice at 7 days after tenotomy. N.S.: not significant. Data represent the mean ± standard deviation. One-way ANOVA, followed by the Tukey–Kramer or Bonferroni test, was used for statistical analyses.

DOI: https://doi.org/10.7554/eLife.48284.017

The following source data and figure supplements are available for figure 5:

**Source data 1.** Numerical data for making the graphs in *Figure 5*.
DOI: https://doi.org/10.7554/eLife.48284.020

**Figure supplement 1.** Number of EdU$^+$ or total myonuclei per 100 myofibers.
DOI: https://doi.org/10.7554/eLife.48284.018

**Figure supplement 1—source data 1.** Numerical data for making the graphs in *Figure 5—figure supplement 1*.
DOI: https://doi.org/10.7554/eLife.48284.019

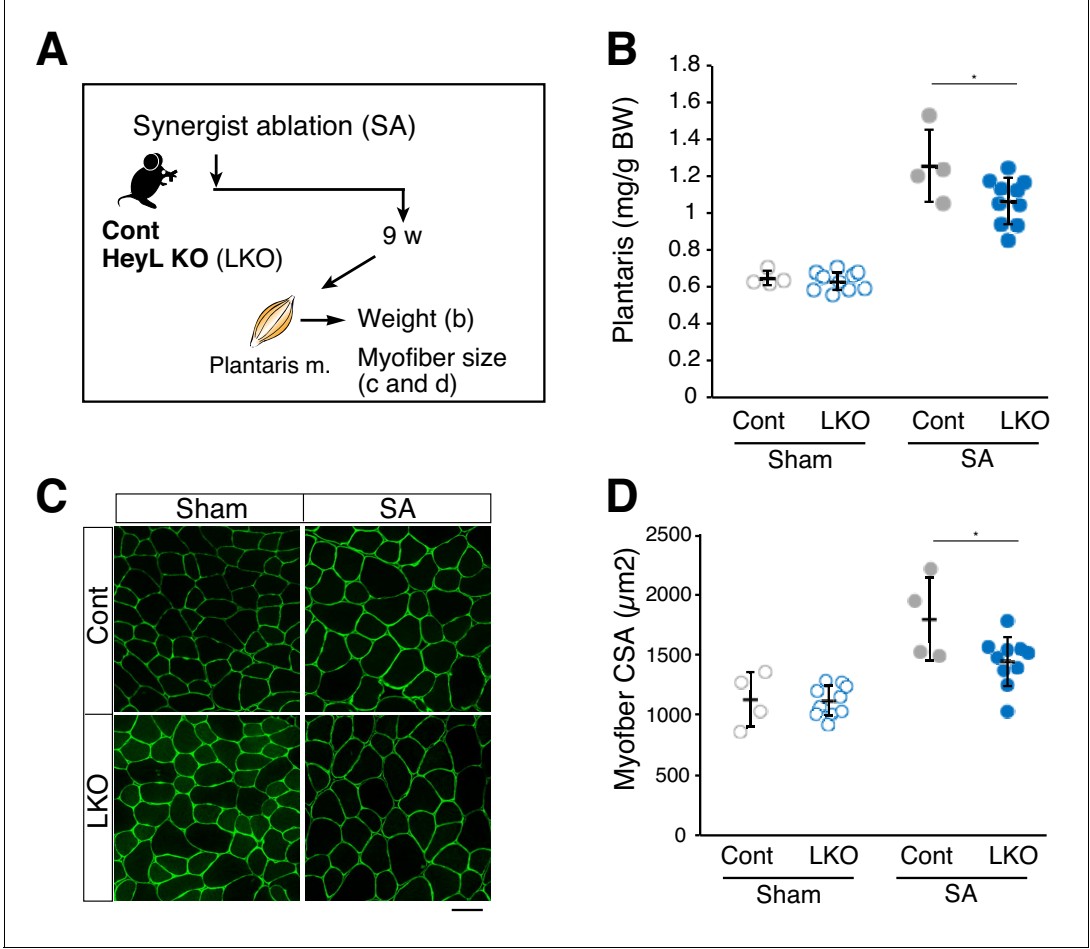

**Figure 6.** Blunted hypertrophy in HeyL-KO mice at 9 weeks after synergist ablation (SA). (A) Experimental scheme for analyzing the influence of HeyL-KO (LKO) on overloaded plantaris muscle at 9 weeks after SA. (B) Normalized plantaris muscle weight [muscle weight/body weight (BW)] from sham (open circle) or SA (closed circle) muscle derived from Cont (light gray, n = 4) or LKO mice (blue, n = 10). *p<0.05. N.S.: not significant. (C) Immunostaining of laminin α2 (green) in sham or SA plantaris muscle derived from Cont or LKO mice. Scale bar: 50 μm. (D) Average myofiber cross-sectional area (CSA) values from sham (open circle) or SA (closed circle) plantaris muscle derived from Cont (light gray, n = 4) or LKO mice (blue, n = 10). *p<0.05. N.S.: not significant. Data represent the mean ± standard deviation. One-way ANOVA, followed by the Tukey–Kramer test, was used for statistical analyses.

DOI: https://doi.org/10.7554/eLife.48284.021

The following source data and figure supplements are available for figure 6:

**Source data 1.** Numerical data for making the graphs in *Figure 6*.
DOI: https://doi.org/10.7554/eLife.48284.024
**Figure supplement 1.** Number of total myonuclei per 100 myofibers 9 weeks after SA.
DOI: https://doi.org/10.7554/eLife.48284.022
**Figure supplement 1—source data 1.** Numerical data for making the graph in *Figure 6—figure supplement 1*.
DOI: https://doi.org/10.7554/eLife.48284.023

Based on our results, the kinetics of MuSC behavior in overloaded muscle also differ from those in regenerating muscles (*Figure 7A*; *Figure 7—figure supplement 1A*). In cardiotoxin (CTX)-induced regenerating muscle, MuSC activation (MyoD expression) occurs quickly, and at least 50% of MuSCs are Ki67[+] at 1 day after injury (*Ogawa et al., 2015*). At 2 to 3 days after injury, MuSCs proliferate vigorously and begin to express myogenin to promote MuSC fusion. In overloaded muscle, MuSC activation (Ki67 expression) is initiated 2 days after tenotomy, followed by MuSC proliferation (*Figure 1*; *Figure 7—figure supplement 1A*). Importantly, proliferating and differentiating MuSCs co-exist for long periods, resulting in gradually increasing numbers of MuSC-derived myonuclei until 14

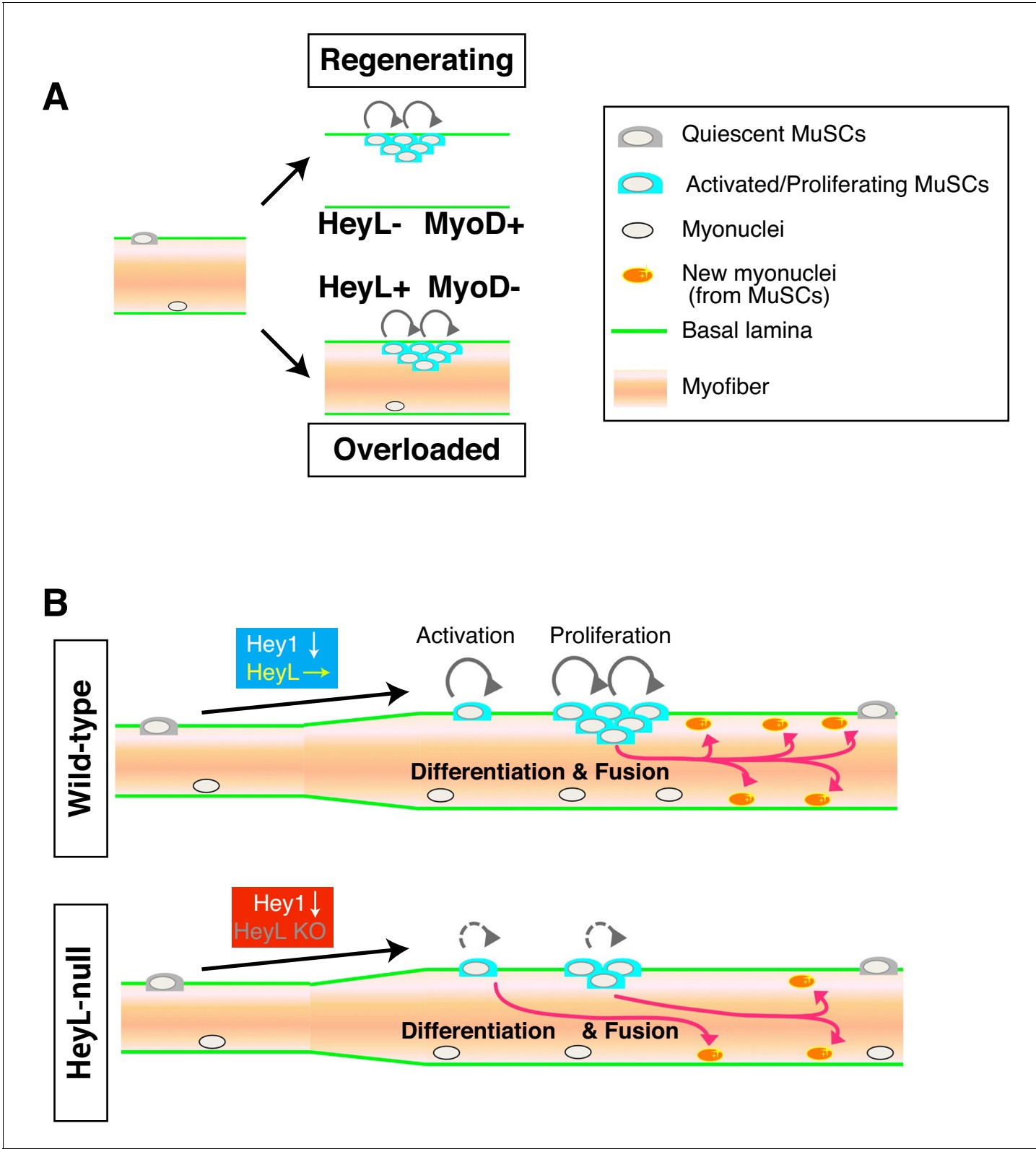

**Figure 7.** Model of MuSC behavior in overloaded muscle. (A) Distinct environments for MuSC proliferation between regenerating and overloaded muscle. (B) Essential roles of HeyL in MuSC proliferation in overloaded muscle.

DOI: https://doi.org/10.7554/eLife.48284.025

The following source data and figure supplements are available for figure 7:

*Figure 7 continued on next page*

*Figure 7 continued*

**Figure supplement 1.** Kinetics of MuSC behavior in overloaded muscle.
DOI: https://doi.org/10.7554/eLife.48284.026
**Figure supplement 1—source data 1.** Raw data in *Figure 7—figure supplement 1*.
DOI: https://doi.org/10.7554/eLife.48284.027

days after tenotomy (*Figure 7—figure supplement 1A–C*). RNA-seq data supported this model based on elevated expression of both cell cycle-related genes and *myogenin* in overloaded MuSCs.

*Hey1* and *Heyl* are effector genes of canonical Notch signaling in MuSCs. In muscle-injury experiments, Hey1-KO and HeyL-KO mice showed normal muscle-regeneration ability (*Fukada et al., 2011*); however, in overloaded muscle, HeyL-KO mice exhibited impaired cell proliferation, which resulted in a decreased number of new myonuclei during muscle hypertrophy (*Figure 7B*). These results strongly support the notion that the MuSC-proliferation mechanism in the overload model differs from that in the injury model. This study did not yield direct evidence indicating active Notch signaling in MuSCs in overloaded muscle, because real-time imaging of Notch signaling is challenging. However, the following results suggested that Notch signaling was activated in proliferating MuSCs of overloaded muscle at 4 days after tenotomy (when MuSCs proliferate actively): 1) in addition to *Heyl*, proliferating MuSCs expressed Notch-related genes and *Col5a1*, *Col5a3*, and *Col6a1*, which are direct target genes associated with canonical Notch signaling (*Baghdadi et al., 2018*); and 2) MuSCs proliferated on myofibers, which are presumed to express Notch ligands. If active Notch signaling is responsible for the sustained expression of *Heyl* via Rbp-J, a key question that remains unanswered is how MuSCs suppress *Hey1* expression. Although RNA-seq data strongly suggested active Notch signaling in the overloaded MuSCs, we cannot exclude the possibility that Notch signaling is inactive in proliferating MuSCs, and that a Notch-independent mechanism induces *Heyl* expression. To the best of our knowledge, this is the first evidence indicating that expression of Notch effector genes is differentially regulated in MuSCs. Elucidation of the detailed mechanisms by which *Hey1* and *Heyl* expression are regulated will promote an in-depth understanding of the mechanism of MuSC proliferation during muscle hypertrophy.

Exercise-induced muscle damage are reportedly involved in skeletal muscle hypertrophy (*Damas et al., 2018*; *Schoenfeld, 2012*). Because MuSCs are indispensable for repairing damaged myofibers (*Lepper et al., 2011*; *Relaix and Zammit, 2012*; *Sambasivan et al., 2011*), MuSCs are likely involved in cases of exercise that produces substantial damage to myofibers. Our experimental methods involved tenotomy, which causes less damage than SA. Additionally, we used isolated single myofibers in order to allow examination of MuSCs on undamaged myofibers. We found that almost all overloaded myofibers harbored proliferating MuSCs rather than quiescent MuSCs, suggesting that even if local injury had occurred, the locally injured areas were not responsible for the MuSC activation and proliferation observed here. Collectively, our results demonstrated that MuSCs could become activated and proliferate and ultimately supply new nuclei, even in environments producing minimal myofiber damage, indicating that substantial damage is unnecessary for MuSC activation/proliferation in hypertrophic muscles.

Our results showed that in overloaded muscle, fewer proliferating MuSCs were present in HeyL-KO mice than in control mice, because the MuSCs were directed toward differentiation; however, the percentage of total Ki67$^+$ cells among MuSCs in HeyL-KO mice was similar to that in control mice, which suggests that MuSC activation was unaffected by the absence of *Heyl*. This result suggested that MuSCs require activation/proliferation-inducing factors; however, the mechanisms underlying MuSC activation from the quiescent state in overloaded muscle remain unclear. Muscle-derived nitric oxide-stimulated secretion of hepatocyte growth factor (HGF) from the extracellular matrix is a potential pathway for MuSC activation/proliferation in overloaded muscle (*Anderson and Pilipowicz, 2002*; *Tatsumi et al., 1998*; *Yamada et al., 2008*). Using mice in which the HGF receptor (c-Met) was specifically deleted in MuSCs, *Webster and Fan (2013)* showed that HGF-specific signaling did not affect MuSC activation/proliferation in an injury model. However, compared with the hypertrophy model, the injury model could potentially involve several complementary factors that contribute to MuSC activation/proliferation; consequently, the contribution of HGF might be obscured in the injury model. Therefore, MuSC-specific c-Met-deleted mice could potentially exhibit defects in MuSC activation and proliferation in overloaded muscle.

Even in overloaded muscle, inflammation is a candidate pathway for inducing MuSC activation and proliferation. *Mackey et al. (2007)* reported that treatment with non-steroidal anti-inflammatory drugs (inhibitors of synthesis of prostaglandins, including PGE2) suppressed MuSC activity in overloaded human muscle. Furthermore, *Ho et al. (2017)* showed that PGE2 promotes MuSC proliferation in an injury mouse model. In the present study, we observed no marked increase in macrophage number at 2 days after tenotomy, when some MuSCs were already activated, which suggests that macrophage infiltration is no required for MuSC activation in overloaded muscle. At 4 days after tenotomy, macrophage number was only slightly increased (data not shown), although their number at this stage is already extremely high in an injury model. Macrophage-depletion models could help ascertain whether MuSCs require inflammation for their proliferation in an undamaged overloaded muscle.

Additionally, a mechanical stimulus could provide insight into a possible pathway for MuSC activation/proliferation in overloaded muscle. In developing myofibers, mechanical-stimulus-dependent Yap1 activation induces the expression of a Notch1 ligand, Jag2, which maintains the pool of myogenic progenitors (*Esteves de Lima et al., 2016*). Whether Yap1 is activated in MuSCs in response to mechanical stimuli remains unclear; however, the possibility exists that mechanical overload directly promotes the transition of MuSCs from the quiescent state and induces their proliferation. Investigation of these possibilities in future studies will support acquisition of a comprehensive understanding of the behaviors of MuSCs in overloaded muscle in which non-lethal damage to myofibers has occurred.

Because the ratio of myofiber nuclear content to myofiber mass remains constant during hypertrophy (known as the myonuclear domain theory), addition of new myonuclei derived from MuSCs is considered essential for increasing muscle mass (*Cheek et al., 1965*). However, using a MuSC-depleted model, *McCarthy et al. (2011)* showed that MuSCs play dispensable roles in muscle hypertrophy at an early stage of the hypertrophy. However, *Egner et al. (2016)* argued against this finding by presenting results demonstrating that plantaris muscle fails to undergo overload-induced hypertrophy in the absence of MuSCs; *Ito et al. (2013)* reported that neuronal nitric oxide synthase was rapidly activated (within 3 min) by overload induced using tenotomy, and that this stimulation eventually triggered the activation of mammalian target of rapamycin signaling, which is a recognized protein-synthesis pathway. In the present study, HeyL-KO mice showed no difference in muscle weight relative to control mice at 1 week after tenotomy, when increased myofiber size was already observed (*Ito et al., 2013*). However, the loss of *Heyl* affected both muscle weight and myofiber size at 9 weeks after SA. Additionally, *Fry et al. (2014)* noted that the absence of MuSCs affected the late stage (8 weeks) of SA-induced muscle hypertrophy. Collectively, the results of the present study indicated that the early stage of muscle hypertrophy does not require MuSCs, but that the addition of new MuSC-derived myonuclei was responsible for long-term increases in myofiber mass.

In summary, our findings reveal an essential mechanism underlying MuSC proliferation during muscle hypertrophy (*Figure 7A*). Identification of factors and mechanism that induce MuSC activation/proliferation under HeyL expression could lead to the development of MuSC-based therapeutic strategies for muscle-wasting diseases.

## Materials and methods

**Key resources table**

| Reagent type (species) or resource | Designation | Source or reference | Identifiers | Additional information |
| --- | --- | --- | --- | --- |
| Strain, strain background (*Mus musculus*) | Hey1-KO | PMID: 15680351 | MGI:3574059 | *Kokubo et al., 2005* |
| Strain, strain background (*Mus musculus*) | Heyl-KO | PMID: 21989910 | MGI:5302514 | *Fukada et al., 2011* |
| Strain, strain background (*Mus musculus*) | Pax7$^{CreERT2/+}$ | The Jackson Laboratory PMID: 19554048 | JAX:# 012476 RRID:IMSR_JAX:012476 | |

*Continued on next page*

Continued

| Reagent type (species) or resource | Designation | Source or reference | Identifiers | Additional information |
|---|---|---|---|---|
| Strain, strain background (*Mus musculus*) | Rosa26$^{EYFP}$ | The Jackson Laboratory PMID: 19554048 | JAX:# 006148 RRID:IMSR_JAX:006148 | |
| Antibody | anti-CD31-FITC (rat monoclonal) | BD Biosciences | Cat# 558738 RRID:AB_397097 | 1:400 |
| Antibody | anti-CD45-FITC (rat monoclonal) | eBioscience | Cat# 11-0451-82 RRID:AB_465050 | 1:800 |
| Antibody | anti-Sca1-PE (rat monoclonal) | BD Biosciences | Cat# 553336 RRID:AB_394792 | 1:400 |
| Antibody | anti-Satellite cells (rat monoclonal) | *Fukada et al., 2004* ECR 296, 245–255. | | 1:200 |
| Antibody | anti-Dystrophin (rabbit polyclonal) | Abcam | Cat# ab15277, RRID:AB_301813 | 1:800 |
| Antibody | anti-eMyHC (mouse monoclonal) | DSHB | Cat# F1.652 RRID:AB_528358 | 1:2 (Supernatant) |
| Antibody | anti-GFP (YFP) (goat polyclonal) | SICGEN | Cat# AB0020-200, RRID:AB_2333099 | 1:1000 |
| Antibody | anti-GFP (YFP) (rabbit polyclonal) | MBL | Cat# 598S, RRID:AB_591816 | 1:1000 |
| Antibody | anti-Ki67 (rat monoclonal) | Thermo Fisher Scientific | Cat# 14-5698-82 RRID:AB_10854564 | 1:200 |
| Antibody | anti-Laminin a2 (rat monoclonal) | Enzo Life Sciences | Cat#ALX-804–190 C100, RRID:AB_2051764 | 1:200 |
| Antibody | anti-M-cadherin (rabbit polyclonal) | *Ojima et al., 2004* | | 1:1000 |
| Antibody | anti-MyoD (rabbit polyclonal) | Santa Cruz | Cat# sc-760 RRID:AB_2148870 | 1:200 |
| Antibody | anti-MyoD (mouse monoclonal) | BD Biosciences | Cat# 554130 RRID:AB_395255 | 1:200 |
| Antibody | anti-Pax7 (mouse monoclonal) | DSHB | Cat# PAX7 RRID:AB_528428 | 1:2 (Supernatant) |
| Commercial assay or kit | Click-iT EdU Cell Proliferation Kit for Imaging, Alexa Fluor 647 dye | Thermo Fisher Scientific | #C10340 | |
| Chemical compound, drug | EdU | Thermo Fisher Scientific | #A10044 | |
| Chemical compound, drug | Cardiotoxin | Latoxan | Cat# L8102 | 10 µM |
| Chemical compound, drug | Tamoxifen | Sigma-Aldrich | Cat# T5648-1G | |
| Others | DAPI | Vector Laboratories | Cat# H-1200 | |
| Others | MOM blocking | Vector Laboratories | Cat# BMK-2202 | |
| Sequence-based reagent | 5'-TGTGGATCACCTGAAAATGC | Eurofins | N/A | qPCR Hey1 (Forward primer) |
| Sequence-based reagent | 5'-ACCCCAAACTCCGATAGTCC | Eurofins | N/A | qPCR Hey1 (Reverse primer) |
| Sequence-based reagent | 5'-CAGCCCTTCGCAGATGCAA | Eurofins | N/A | qPCR Heyl (Forward primer) |
| Sequence-based reagent | 5'-CCAATCGTCGCAATTCAGAAAG | Eurofins | N/A | qPCR Heyl (Reverse primer) |

*Continued on next page*

*Continued*

| Reagent type (species) or resource | Designation | Source or reference | Identifiers | Additional information |
|---|---|---|---|---|
| Sequence-based reagent | 5'-GCTACTCCTG TTCCTGCTGC | Eurofins | N/A | qPCR Col5a1 (Forward primer) |
| Sequence-based reagent | 5'-TGAGGGCAA ATTGTGAAAATC | Eurofins | N/A | qPCR Col5a1 (Reverse primer) |
| Sequence-based reagent | 5'-CCGGAGACTGG ATCAGCTT | Eurofins | N/A | qPCR Col5a3 (Forward primer) |
| Sequence-based reagent | 5'-GCTTCCAGTAC GTCCACAGG | Eurofins | N/A | qPCR Col5a3 (Reverse primer) |
| Sequence-based reagent | 5'-TACTTCGGGAAA GGCACCTA | Eurofins | N/A | qPCR Col6a1 (Forward primer) |
| Sequence-based reagent | 5'-TCGGTCACC ACGATCAAGT | Eurofins | N/A | qPCR Col6a1 (Reverse primer) |
| Chemical compound, drug | Trizol LS reagent | Thermo Fisher Scientific | Cat# 10296028 | |
| Chemical compound, drug | RNeasy mini kit | QIAGEN | Cat# 74104 | |
| Software, algorithm | Photophop CC | https://www.adobe.com/products/photoshop.html | RRID:SCR_014199 | |
| Software, algorithm | Hybrid Cell count | https://www.keyence.com/ss/products/microscope/bz-x/products-info/quantify.jsp | | |
| Software, algorithm | UMI-tools | https://github.com/CGATOxford/UMI-tools | | v.1.1.0 |
| Software, algorithm | HISAT2 | https://ccb.jhu.edu/software/hisat2/index.shtml | | v.2.1.0 |
| Software, algorithm | featureCounts | http://subread.sourceforge.net/ | | v.1.6.4 |
| Software, algorithm | DESeq2 | https://bioconductor.org/packages/release/bioc/html/DESeq2.html | DOI: 10.18129/B9.bioc.DESeq2 | v.1.22.2 |

## Mice

C57BL/6J mice were purchased from Charles River Laboratories (Yokohama, Kanagawa, Japan). *Hey1*$^{-/-}$ mice and *Heyl*$^{-/-}$ mice were described previously (*Fukada et al., 2011*; *Kokubo et al., 2005*). *Pax7*$^{CreERT2+}$ mice (stock No: 012476) were obtained from Jackson Laboratories (Bar Harbor, ME, USA; Farmington, CT, USA; or Sacramento, CA, USA). Relocation of *Rosa26*$^{EYFP/+}$ mice (stock No: 006148) from the National Center of Neurology and Psychiatry to Osaka University was approved by Jackson Laboratories. *Pax7*$^{CreERT2+}$::*Rosa26*$^{EYFP/+}$ mice were injected intraperitoneally twice (24 hr apart) with 200 µl to 300 µl tamoxifen (20 mg/ml; #T5648; Sigma-Aldrich, St. Louis, MO, USA) dissolved in sunflower seed oil (#S5007; Sigma-Aldrich) and 5% ethanol, and the mice were maintained in a controlled environment (temperature, 24 ± 2°C; humidity, 50 ± 10%) under a 12/12 hr light/dark cycle. The mice received sterilized standard chow (DC-8; Nihon Clea, Tokyo, Japan) and water *ad libitum*. All procedures used for the experimental animals were approved by the Experimental Animal Care and Use Committee of Osaka University (approval number: 25-9-3, 30–15).

## Muscle-overload model

Mice were anesthetized with isoflurane. Functional overloading of plantaris muscle was induced as described in a previous report (*Ito et al., 2013*). Briefly, a midline incision was made in the skin on the hindlimbs, and the distal tendons of both gastrocnemius and soleus muscles were transected (tenotomy). In the experiments shown in *Figure 6*, the overloading of plantaris muscle was induced by excising the distal half of both the gastrocnemius and the soleus (i.e., SA was performed). The

incision was closed using a 7–0 silk suture (Matsuda Ika Kogyo Co., Ltd., Tokyo, Japan). For the sham-operated group, similar incisions were made in the skin, but the tendons were not transected. Muscle weights were normalized against body weights.

## Muscle-injury model

Muscle regeneration was induced by injecting 10 µM CTX (Latoxan, Valence, France) in phosphate-buffered saline (PBS) into the tibialis anterior muscle (50 µl).

## Muscle fixation and histologic analysis

Isolated plantaris or tibialis anterior muscles were frozen in liquid-nitrogen-cooled isopentane (Wako Pure Chemicals Industries, Osaka, Japan) for 1 min and then placed on dry ice for 1 hr to vaporize the isopentane. Transverse cryosections (10 µm) were stained with hematoxylin and eosin (H and E).

For immunohistochemical analyses, transverse cryosections (6 µm) were fixed with 4% paraformal-dehyde (PFA) for 10 min. For embryonic myosin heavy chain (eMyHC) staining, sections were fixed with cooled acetone for 10 min at −20°C. Detailed information on antibodies used in this study is provided in Key Resource Table. For mouse anti-Pax7 and -eMyHC staining, a M.O.M. kit (Vector Laboratories, Burlingame, VT, USA) was used to block endogenous mouse IgG before reacting with the primary antibodies. The signals were recorded photographically using a BZ-X700 fluorescence microscope (Keyence, Osaka, Japan).

## Single-myofiber isolation, culture, and staining

Single myofibers were isolated from plantaris muscles according to a previously described protocol (*Rosenblatt et al., 1995*). The isolated myofibers were immediately fixed in 4% PFA or cultured in high-glucose Dulbecco's modified Eagle medium containing 10% fetal calf serum. Immunostaining was performed using previously described protocols (*Collins-Hooper et al., 2012*; *Shinin et al., 2009*), and the stained myofibers were recorded using a BZ-X700 fluorescence microscope (Keyence).

## In vivo EdU labeling and detection

EdU (#A10044; Thermo Fisher Scientific, Waltham, MA, USA) was dissolved in PBS at 0.5 mg/ml and injected intraperitoneally into mice at 0.1 mg per 20 g body weight (5 mg/kg) daily until the day before sacrifice. EdU labeling was detected using the protocol supplied by the manufacturer (#C10340; Thermo Fisher Scientific).

## Isolation of skeletal muscle-derived mononuclear cells

Sham or overloaded plantaris muscles or uninjured/injured limb muscles were digested using 0.2% collagenase type II (Worthington Biochemical, Lakewood, NJ, USA). Mononuclear cells derived from skeletal muscle were stained with FITC-conjugated anti-CD31 and anti-CD45, PE-conjugated anti-Sca-1, and biotinylated-SM/C-2.6 antibodies (*Fukada et al., 2004*). Subsequently, the cells were incubated with streptavidin-labeled allophycocyanin (BD Biosciences, San Jose, CA, USA) on ice for 30 min and resuspended in PBS containing 2% fetal bovine serum and 2 mg/ml propidium iodide. Cell sorting was performed using a FACS Aria II flow cytometer (BD Immunocytometry Systems, San Jose, CA, USA). Muscle stem cells from $Pax7^{CreERT2+}::Rosa^{YFP}$ mice were sorted by yellow fluorescent protein fluorescence (YFP) for RNA-seq analyses.

## RNA isolation and qRT-PCR analysis

Total RNA was extracted from sorted cells by using Trizol LS reagent (Thermo Fisher Scientific) and a QIAGEN RNeasy mini kit according to the manufacturer instructions (QIAGEN, Hilden, Germany) and then reverse-transcribed into cDNA using a QuantiTect reverse transcription kit (QIAGEN). Specific forward and reverse primers used for optimal amplification during real-time PCR of reverse-transcribed cDNAs are listed in Key Resource Table.

## CEL-Seq2 and data analysis

Cells were sorted into tubes containing lysis buffer at 100 cells/tube using a cell sorter (Aria II). Libraries were constructed according to a previously described protocol (*Hashimshony et al., 2016*)

and sequenced using an Illumina HiSeq 1500 system (Illumina, San Diego, CA, USA). The barcodes and unique-molecular-identifiers in the reads were extracted using UMI-tools (v.1.0.0) (*Smith et al., 2017*) with the following command: 'umi_tools extract -I r1.fastq -read2-in = r2.fastq -bc-pattern=NNNNNNCCCCCC -read2-stdout'. The reads were mapped to the reference genome (GRCm38) using the alignment software HISAT2 (v.2.1.0) (*Kim et al., 2015*). Read counts per gene were determined using featureCounts (v.1.6.4) (*Liao et al., 2014*) and UMI-tools using the following command: 'umi_tools count -method=directional per-gene -per-cell -gene-tag=XT'. Duplicates among unique molecular identifiers were discarded. Differentially expressed genes (adjusted p<0.1) were extracted using the R library program DESeq2 (v.1.22.2) (*Love et al., 2014*). Data were deposited with the accession number GSE135903. Gene-expression data for myogenic cells at 2 days after CTX injection are available under accession number GSE56903 (*Ogawa et al., 2015*).

## Statistics

Values are expressed as the mean ± standard deviation Statistical significance was assessed using the F test, followed by Student's *t* test. When $p<0.05$ for the F test, Welch's *t* test was used. For comparisons more than two groups, we used non-repeated measures analysis of variance (ANOVA), followed by the Tukey–Kramer test. When $p \geq 0.05$ for the ANOVA results, a Bonferroni test (vs. control) was used (*Figure 5D*). A $p<0.05$ or $p<0.01$ was considered statistically significant.

## Acknowledgements

S-iF was funded by a Grant-in-Aid for Scientific Research (B), an Intramural Research Grant for Neurological and Psychiatric Disorders of NCNP, the Naito Foundation, Nakatomi Foundation, and the Ichiro Kanehara Foundation. This research was also supported by AMED under grant no 18am0101084j0002.

## Additional information

### Competing interests

Sumiaki Fukuda, Akihiro Kaneshige: Employee of Japan Tobacco Inc at the time the study was conducted. There are no other competing financial interests to declare. The other authors declare that no competing interests exist.

### Funding

| Funder | Grant reference number | Author |
|---|---|---|
| Ministry of Education, Culture, Sports, Science, and Technology | Grant-in-Aid for Scientific Research (B) | So-ichiro Fukada |
| Naito Foundation | | So-ichiro Fukada |
| Nakatomi Foundation | | So-ichiro Fukada |
| Ichiro Kanehara Foundation for the Promotion of Medical Sciences and Medical Care | | So-ichiro Fukada |
| National Center of Neurology and Psychiatry | Intramural Research Grant for Neurological and Psychiatric Disorders | So-ichiro Fukada |
| Japan Agency for Medical Research and Development | 18am0101084j | Kazutake Tsujikawa |

The funders had no role in study design, data collection and interpretation, or the decision to submit the work for publication.

## Author contributions

Sumiaki Fukuda, Data curation, Formal analysis, Validation, Investigation, Writing—original draft; Akihiro Kaneshige, Akihito Harada, Data curation, Investigation; Takayuki Kaji, Yu-taro Noguchi, Yusei Takemoto, Lidan Zhang, Investigation; Kazutake Tsujikawa, Resources, Funding acquisition; Hiroki Kokubo, Resources; Akiyoshi Uezumi, Conceptualization, Resources; Kazumitsu Maehara, Data curation, Software, Investigation; Yasuyuki Ohkawa, Resources, Software, Methodology; So-ichiro Fukada, Supervision, Funding acquisition, Writing—original draft, Project administration, Writing—review and editing

## Author ORCIDs

So-ichiro Fukada https://orcid.org/0000-0003-4051-5108

## Ethics

Animal experimentation: All procedures used for experimental animals were approved by the Experimental Animal Care and Use Committee of Osaka University (approval number: 25-9-3, 30-15).

## Decision letter and Author response

Decision letter https://doi.org/10.7554/eLife.48284.030
Author response https://doi.org/10.7554/eLife.48284.031

# Additional files

## Supplementary files

• Transparent reporting form
DOI: https://doi.org/10.7554/eLife.48284.028

## Data availability

All data generated or analyzed in this study are included in the manuscript and supporting files.

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
