## [Decision Letter]

Thank you for submitting your article "Different requirement of *HeyL* in proliferating muscle stem cells in overloaded versus regenerating muscle" consideration by *eLife*. Your article has been reviewed by three peer reviewers, including a guest Reviewing Editor, and the evaluation has been overseen by Didier Stainier as Senior Editor. The reviewers have opted to remain anonymous.

The reviewers have discussed the reviews with one another and the Reviewing Editor has drafted this decision to help you prepare a revised submission.

Summary:

This is an interesting manuscript that shows that the mode of muscle stem cell activation differs between cardiotoxin-mediated injury and overloading-induced hypertrophy. It remains an open question how much from injury-model can be applied to more physiological settings. Unexpectedly, the authors find different features in the overloading model – specifically, the stem cells are activated / proliferate / fuse while *MyoD* is weakly induced. Mechanistically, one of the Notch pathway targets, *HeyL*, is sustained in overload model, which in turn might suppress *MyoD*. Loss of *HeyL* decreases stem cell activation and fusion in the overload model, and the authors interpret that accumulation of *MyoD* forces the stem cells to differentiate. Overall, this is a potentially interesting paper trying to address an important question. However, there are some unclear points and potentially alternative explanations about what the authors are observing. Additional experiments are needed in order to substantiate the conclusions.

Major concern:

The interpretation that *HeyL* KO mouse fails to induce *MyoD* is ambiguous. In CTX injury model, both *HeyL*/Hey1 goes down leading to an stronger *MyoD* induction. An alternative explanation is that all the pathways are just less strongly activated in the overload model compared to injury – not necessarily that there is a completely different mechanism of activation. Indeed, the extent of stem cell number increase is small in overload compared to injury (Figure 1D). This could be the reason why *MyoD* induction is also weaker. Further, Hey1 might be just a less efficient target of Rbpj, and that could be the reason why only Hey1 is down-regulated while *HeyL* is sustained. In this point of view, it would be much more convincing if the authors could perform unbiased transcriptome analysis of the stem cells in injury versus overload, and show that there are actually a group of genes that goes up in overload model.

Overall, this manuscript is well written, interesting and timely and might help find a new mode of activation of satellite cells in the absence of aggressive muscle damage such as cardiotoxin-induced injury. We have the following suggestions to improve the manuscript.

Essential revisions:

1) The authors mentioned that *MyoD*- MuSCs undergo cell proliferation during muscle overloading within the intact myofibers. However, this data is still missing. Please show *MyoD*, EdU and laminin staining, and possibly in combination with Pax7 in overloaded muscle sections. Especially, the authors should analyze time course-dependent kinetics of MuSC activation, entering cell cycle and exiting cell cycle, and fusion to intact myofibers during muscle overloading. This data will confirm that MuSC located underneath the basal lamina can enter cell cycle and contribute to myonuclei during overloading. Similar immunodetection should be performed for overloaded muscle sections in *HeyL* KO mice.

2) "importantly, at 4 days after tenotomy, the percentage of Pax7+Ki67+ cells among Pax7+ cells (~70%) was higher than that of Pax7+*MyoD*+ cells (~35%), which suggested that most of these MuSCs proliferated in the absence of *MyoD* expression." This statement is a gross oversimplification, since it is unclear whether the majority of Pax7+Ki67+ cells are proliferating. Maybe only half are cycling?

3) Subsection “Proliferation of MuSCs during muscle hypertrophy”, last paragraph: These results suggested that MuSC proliferation does not occur in a restricted area of the overloaded myofibers and that almost all myofibers respond to the overload, which awakens MuSCs from quiescence. Again the authors use Ki67 expression as synonym with proliferation- which it is not. It is an indication that cells are not quiescent.

4) In Figure 3, *Hes1* expression should be included since MuSC can express *Hes1* according to the publication (Lahmann et al., 2019).

5) Changes in gene expression in the overload model. Only *Hey/HeyL* and a couple of collagen genes are tested. Please provide a rational for selecting these collagens. The authors should analyze a broader set of Notch target genes and other myogenic markers. Best would be to provide RNAseq data, or at least a broad set of Notch target genes including more members of the Hes/Hey family, Notch ligands and receptors, *MyoD* and other myogenic transcription factors (*Myf5* etc.). Please provide SEM / error bars for experiment Figure 3B and C.

6) The finding suggests that MuSCs proliferate under activation of canonical Notch signaling in the overload mouse model and that MuSCs lose Hey1 expression despite Notch signaling being active. This is an over-interpretation of the scarce data provided. The fact that one gene (*HeyL*) is upregulated provides no evidence that Notch signaling is active. The authors could equally argue that Notch signaling is absent since Hey1 is downregulated.

7) Figure 4: The effects of the *HeyL* mutations on Ki67 and *MyoD* expression are mild; the ratio of *MyoD*+/*MyoD*- cells is larger but extremely variable; it would be nice if n could be increased in this experiment.

8) Subsection “Supply of new myonuclei is diminished in *HeyL*-KO-mouse overloaded muscle”: Approximately 20 or 60 myonuclei were labeled per cross-section from control/Hey1-KO mice at 4 or 7 days after surgery. The quantification of this is unclear since the muscle diameter might be highly variable depending on the section: please quantify not per cross-section but per myofiber or per 100 myofiber nuclei.

9) Figure 5B – In the *HeyL* KO image, there are a lot of EdU positive nuclei that are not assigned as myonuclei. In contrast, in control or Hey1 KO, all EdU nuclei are assigned as myonuclei. What are the non-myonuclei/EdU labeled nuclei in the *HeyL* KO? Are they Pax7 positive?

---

## [Author Response]

Major concern:The interpretation that HeyL KO mouse fails to induce MyoD is ambiguous. In CTX injury model, both HeyL/Hey1 goes down leading to an stronger MyoD induction. An alternative explanation is that all the pathways are just less strongly activated in the overload model compared to injury – not necessarily that there is a completely different mechanism of activation. Indeed, the extent of stem cell number increase is small in overload compared to injury (Figure 1D). This could be the reason why MyoD induction is also weaker. Further, Hey1 might be just a less efficient target of Rbpj, and that could be the reason why only Hey1 is down-regulated while HeyL is sustained. In this point of view, it would be much more convincing if the authors could perform unbiased transcriptome analysis of the stem cells in injury versus overload, and show that there are actually a group of genes that goes up in overload model.

We thank the reviewers for this careful reading of our manuscript. In myogenic cells, we and other groups previously showed *HeyL* as a critical and direct target of Notch signaling. In the present study, our data indicated a modest increase in the number of satellite cells due to myogenic differentiation (fusion with myofibers), with proliferation simultaneously observed in overloaded muscle. *HeyL* KO failed to suppress *MyoD* expression in overloaded and resulted in no associated phenotype in regenerating muscles, indicating that the sustained expression of *HeyL* is a feature of and important to overloaded MuSCs.

According to the reviewers’ suggestion, we added the results of transcriptome analyses of overloaded MuSCs to revised Figure 3.

Overall, this manuscript is well written, interesting and timely and might help find a new mode of activation of satellite cells in the absence of aggressive muscle damage such as cardiotoxin-induced injury. We have the following suggestions to improve the manuscript.

We are thankful for the recognition of the value of our work. We are confident that the findings will promote an in-depth understanding of MuSC behavior in overloaded muscle.

Essential revisions:1) The authors mentioned that MyoD- MuSCs undergo cell proliferation during muscle overloading within the intact myofibers. However, this data is still missing. Please show MyoD, EdU and laminin staining, and possibly in combination with Pax7 in overloaded muscle sections. Especially, the authors should analyze time course-dependent kinetics of MuSC activation, entering cell cycle and exiting cell cycle, and fusion to intact myofibers during muscle overloading. This data will confirm that MuSC located underneath the basal lamina can enter cell cycle and contribute to myonuclei during overloading. Similar immunodetection should be performed for overloaded muscle sections in HeyL KO mice.

We appreciate the reviewers’ constructive feedback. Since *MyoD* staining in sections is sometimes difficult to judge, we mainly analyzed MuSCs with single myofiber technique. In revised Figure 3, we added results indicating proliferating Pax7^+^ cells beneath the basal lamina. To describe the kinetics of MuSC behavior, we added the following information obtained during muscle overload:

During activation (day 2), ~10% of MuSCs were Ki67^+^ (Figure 1E), and EdU^+^Pax7^+^ cells were not detected (Figure 7—figure supplement 1B), indicating that MuSCs were initially activated at day 2 after tenotomy.

During MuSC proliferation (days 4–7), ~50% to 70% and a half of MuSC were Ki67^+^ (Figure 1E) and EdU^+^, respectively (Figure 7—figure supplement 1B), indicating that MuSCs proliferated between 4 and 7 days after tenotomy.

Upon exit of cell cycle/fusion (days 4–14), EdU^+^ myonuclei were detected at 4 days after tenotomy, and this number gradually increased until 14 days after tenotomy (Figure 5C and Figure 7—figure supplement 1C). These results indicated that MuSCs exited from the cell cycle and fused with myofiber over a long period of time. Results from *HeyL*-KO mice are shown in Figure 5C.

Therefore, compared to regenerating muscle, vigorous proliferation and differentiation occurred simultaneously in overloading muscle. This result explains the attenuated number of MuSCs in overloaded muscle based on the decreased accumulation of proliferating MuSCs. Furthermore RNA-seq data indicated simultaneous expression of cell cycle-related genes and *myogenin*.

At 14 days after tenotomy, we rarely detected Ki67^+^ MuSCs (Figure 1E). Additionally, the number of MuSCs was similar to that in sham muscle (Figure 1D), indicating that these MuSCs returned to a quiescent state at 14 days after tenotomy.

Moreover, all Pax7^+^EdU^+^ cells were located beneath the basal lamina, including in *HeyL*-KO mice. The frequency of Ki67^+^ cells in *HeyL*-KO mice was similar to that in control mice. On the other hand, the number of MuSCs was reduced in *HeyL*-KO mice (Figure 4E), indicating that the reduced number of new myonuclei resulted from impaired MuSC proliferation but not activation.

The following was added to the revised manuscript:

“Based on our results, the kinetics of MuSC behavior in overloaded muscle also differ from those in regenerating muscles (Figure 7A; Figure 7—figure supplement 1A). —figure supplement[…] RNA-seq data supported this model based on elevated expression of both cell cycle-related genes and *myogenin* in overloaded MuSCs.”

2) "importantly, at 4 days after tenotomy, the percentage of Pax7+Ki67+ cells among Pax7+ cells (~70%) was higher than that of Pax7+MyoD+ cells (~35%), which suggested that most of these MuSCs proliferated in the absence of MyoD expression." This statement is a gross oversimplification, since it is unclear whether the majority of Pax7+Ki67+ cells are proliferating. Maybe only half are cycling?

We thank the reviewers for this comment. The text was changed as follows:

“…suggesting that at least >50% of these MuSCs proliferated in the absence of *MyoD* expression (Figure 1E, F).”

3) Subsection “Proliferation of MuSCs during muscle hypertrophy”, last paragraph: These results suggested that MuSC proliferation does not occur in a restricted area of the overloaded myofibers and that almost all myofibers respond to the overload, which awakens MuSCs from quiescence. Again the authors use Ki67 expression as synonym with proliferation- which it is not. It is an indication that cells are not quiescent.

We thank the reviewer for this important comment. We agree that Ki67 expression does not independently indicate cell proliferation. We changed the text as follows:

“These results suggested that escape of MuSCs from quiescent state did not occur in a restricted area of the overloaded myofibers, and that almost all myofibers responded to the overload.”

4) In Figure 3, Hes1 expression should be included since MuSC can express Hes1 according to the publication (Lahmann et al., 2019).

As previously noted, we performed RNA-seq analyses, with the results showing *Hes1* expression indicated in revised Figure 3. The importance of *Hes1* and the *Hes1*–*HeyL* heterodimer were mentioned in the revised manuscript.

“*Hey1, HeyL*, and *Hes1* are crucial primary targets of canonical Notch signaling in MuSCs (Fukada et al., 2011; Lahmann et al., 2019) and function to suppress myogenic differentiation as *Hey*–*Hes1* heterodimer complexes (Noguchi et al., 2019).”

5) Changes in gene expression in the overload model. Only Hey/HeyL and a couple of collagen genes are tested. Please provide a rational for selecting these collagens. The authors should analyze a broader set of Notch target genes and other myogenic markers. Best would be to provide RNAseq data, or at least a broad set of Notch target genes including more members of the Hes/Hey family, Notch ligands and receptors, MyoD and other myogenic transcription factors (Myf5 etc.). Please provide SEM / error bars for experiment Figure 3B and C.

According to the reviewers’ comment, we performed RNA-seq analyses and obtained similar results as those obtained from qRT-PCR. The sustained expression of *HeyL* in overloaded MuSCs represents the biggest difference in gene-expression pattern from that for regenerating MuSCs. The expression of Notch target genes and myogenic genes have also been provided in revised Figure 3. These results suggested active Notch signaling in overloaded MuSCs. Moreover, elevated *myogenin* expression suggested the existence of differentiated cells at 4 days after tenotomy (see our response to comment 1). Compared to regenerating MuSCs, the sustained expression of *Myf5* was a new finding.

We could obtain only 1000 MuSCs per one sham plantaris muscle. For qRT-PCR analyses, we used 17 heads to obtain sufficient MuSCs from plantaris muscles. Therefore, there was a substantial limitation of those analyses. We repeated the qRT-PCR analysis using the same samples and obtained same results. Therefore, our conclusion was based on qRT-PCR results.

6) The finding suggests that MuSCs proliferate under activation of canonical Notch signaling in the overload mouse model and that MuSCs lose Hey1 expression despite Notch signaling being active. This is an over-interpretation of the scarce data provided. The fact that one gene (HeyL) is upregulated provides no evidence that Notch signaling is active. The authors could equally argue that Notch signaling is absent since Hey1 is downregulated.

We added the following information to the revised manuscript:

“Although RNA-seq data strongly suggested active Notch signaling in the overloaded MuSCs, we cannot exclude the possibility that Notch signaling is inactive in proliferating MuSCs, and that a Notch-independent mechanism induces *Heyl* expression.”

7) Figure 4: The effects of the HeyL mutations on Ki67 and MyoD expression are mild; the ratio of MyoD+/MyoD- cells is larger but extremely variable; it would be nice if n could be increased in this experiment.

We added this information to the revised manuscript accordingly [control (*n* = 1) and *HeyL*-KO (*n* = 2)].

8) Subsection “Supply of new myonuclei is diminished in HeyL-KO-mouse overloaded muscle”: Approximately 20 or 60 myonuclei were labeled per cross-section from control/Hey1-KO mice at 4 or 7 days after surgery. The quantification of this is unclear since the muscle diameter might be highly variable depending on the section: please quantify not per cross-section but per myofiber or per 100 myofiber nuclei.

According to the reviewers’ suggestion, we added data indicating EdU^+^ myonuclei per 100 myofibers (Figure 5—figure supplement 1A). These data indicated significant decreases in the number of EdU^+^ cells per 100 myofibers in *HeyL*-KO mice.

9) Figure 5B – In the HeyL KO image, there are a lot of EdU positive nuclei that are not assigned as myonuclei. In contrast, in control or Hey1 KO, all EdU nuclei are assigned as myonuclei. What are the non-myonuclei/EdU labeled nuclei in the HeyL KO? Are they Pax7 positive?

We apologize for the confusion regarding the *HeyL* results. EdU^+^ cells outside of the dystrophin-staining area represent satellite cells, mesenchymal progenitors, and hematopoietic cells. To avoid confusing readers on this point, the original images were replaced with new ones. Additionally, we did not observe clear Pax7^+^EdU^+^ cells outside of the basal lamina. Now, we have investigated the roles of interstitial cells during muscle hypertrophy and will answer the role of interstitial EdU^+^ cells in future study.